# Improved Sensitivity of α-Fe_2_O_3_ Nanoparticle-Decorated ZnO Nanowire Gas Sensor for CO

**DOI:** 10.3390/s19081903

**Published:** 2019-04-22

**Authors:** Jeongseok Lee, Se-Hyeong Lee, So-Young Bak, Yoojong Kim, Kyoungwan Woo, Sanghyun Lee, Yooseong Lim, Moonsuk Yi

**Affiliations:** 1Department of Electronics Engineering, Pusan National University, Busan 46241, Korea; leejs6015@naver.com (J.L.); shlee12@pusan.ac.kr (S.-H.L.); bso6459027@pusan.ac.kr (S.-Y.B.); yoojongkim@pusan.ac.kr (Y.K.); kwwoo200@pusan.ac.kr (K.W.); lys9003@pusan.ac.kr (Y.L.); 2Department of Smart Hybrid Engineering, Pusan National University, Busan 46241, Korea; collie20@pusan.ac.kr

**Keywords:** gas sensors, nanowires, n–n heterojunction, vapor–liquid–solid, sol–gel, ZnO

## Abstract

A strategy for improving the sensitivity of a sensor for detecting CO and NH_3_ gases is presented herein. The gas sensor was fabricated from ZnO metal oxide semiconductor nanostructures grown via a vapor–liquid–solid process and decorated with α-Fe_2_O_3_ nanoparticles via a sol–gel process. The response was enhanced by the formation of an α-Fe_2_O_3_/ZnO n–n heterojunction and the growth of thinner wires. ZnO nanowires were grown on indium–tin–oxide glass electrodes using Sn as a catalyst for growth instead of Au. The structure and elemental composition were investigated using field-emission scanning electron microscopy, energy dispersive X-ray spectroscopy, and X-ray diffraction. The gas sensing results indicate that the response value to 100 ppm CO was 18.8 at the optimum operating temperature of 300 °C.

## 1. Introduction

ZnO exhibits high electron mobility and is an n-type semiconductor with a wide bandgap (E_g_ = 3.37 eV). α-Fe_2_O_3_ is an environmentally friendly n-type semiconductor with a bandgap of E_g_ = 2.2 eV and is the most stable iron oxide on earth. These materials have garnered significant attention in various fields and are particularly suitable for gas sensors because they react well with harmful gases, such as NO_2_, CO, and H_2_ [1,2,3,4].

Metal oxide semiconductor (MOS)-type gas sensors exhibit advantages of a simple structure, high sensitivity, and low cost [5,6]. In an n-type semiconductor, oxygen vacancies typically form in MOSs. To complement oxygen vacancies, oxygen ions (O^−^, O^2−^) were attached to the surface of an MOS by depriving the MOS of electrons from 200 to 400 °C [7]. Therefore, an electrical core–shell structure was formed, where the shell is an electronic depletion layer at the surface, and the core is an undepleted semiconducting metal oxide layer. Absorbed oxygen oxidizes or reduces the reaction gas. This causes the MOS to lose or trap electrons and changes the thickness of the electronic depletion layer, thus changing the resistance. In other words, electrons from absorbed oxygen are injected into the semiconductor in response to reducing gases. This decreases the thickness of the depletion layer at the surface and the resistance (Figure 1). The change in resistance is proportional to the concentration of exposed gas and is used as the response signal of the gas sensor [8,9,10,11].

MOS-type gas sensors manufactured using only a single material exhibit limited sensitivity and selectivity. This limitation was overcome by forming a hierarchical structure or decorating the structure with materials such as Au, Co, or CuO. These materials have proven to be promising for enhancing the response of these sensors to various gases [11,12,13,14,15,16].

In this study, ZnO nanowires (NWs) were grown on indium tin oxide (ITO) and decorated with α-Fe_2_O_3_ nanoparticles. These structures were used as a gas sensor. Sn acted as a catalyst for the growth of ZnO nanostructures, and the process costs less than using Au as a catalyst for growth [17]. The n–n heterojunction between α-Fe_2_O_3_ and ZnO nanostructures improved the sensitivity to reducing gases.

## 2. Materials and Methods

### 2.1. Growth of ZnO Nanostructures

ZnO NWs were grown using a tube furnace (Figure 2). 

NW growth via vapor liquid solid (VLS) process proceeds as follows. First, the catalyst for growth is deposited in the form of liquid droplets on the substrate. The gas source material subsequently enters the liquid catalyst. When the gas saturates in the droplets, crystalline NWs grow downward from the droplet surface (Figure 3a). In this experiment, ZnO NWs were grown on glass substrates with interdigitated ITO electrodes using a mixture of Zn (>150 μm, 99.995%, Sigma–Aldrich, Korea) and Sn (100 mesh, 99.99%, Sigma–Aldrich, Korea) powders. Sn in ITO and Sn in the powder acted as a catalyst for the growth of ZnO NWs. A mixture of powders was used because the Sn powder created a smaller droplet, thus yielding a thinner wire [18]. With the addition of Sn powder, the NWs grew to a smaller diameter and tangled and bent, resembling vines. Thus, we call the nanostructures NW vines (Figure 3b). The alumina boat containing the mixed powders was placed in the center of the quartz tube, and the side facing the center of the furnace was removed from the alumina boat. The ITO substrates were placed on an alumina boat 18 to 20 cm downstream of the source material. Subsequently, the pressure of the quartz tube was maintained at low vacuum (2 × 10^−3^ Torr) by mechanical pumping to remove impurities. Next, the temperature of the tube furnace was increased to 900 °C while purging with 100-ppm N_2_ gas for 40 min using a mass flow controller. The ZnO NWs were grown at 900 °C for 1 h in an N_2_–O_2_ gas mixture (100 sccm N_2_, 1 sccm O_2_) at 1 Torr, causing the mixed powder (Zn:Sn = 1:1 wt%) to vaporize. Sn gas became liquefied on the substrate because the temperature decreased abruptly from the outside of the heat-resisting wall. However, Zn did not liquefy because its boiling point is much lower than that of Sn. 

### 2.2. Synthesis of α-Fe_2_O_3_/ZnO Heterojunction

Iron (III) chloride hexahydrate (FeCl_3_∙6H_2_O, ≥ 98%, Sigma–Aldrich Korea) was dissolved in ethanol to synthesize an α-Fe_2_O_3_ solution. This solution was stirred for 1 h at 500 rpm and 60 °C. The solution was filtered with a 0.2-μm syringe filter. This solution was deposited on the ZnO nanostructures by spin coating at 3000 rpm for 30 s. The substrates were placed in a tube furnace to heat the solvent for 1 h in an N_2_ atmosphere at 400 °C. The gel grew into nanoparticles (Figure 4). In this experiment, the molar concentration of the iron (III) oxide solution added to the gas sensor was 0, 0.025, 0.05, 0.1, and 0.15 M. The following chain of chemical reactions produced Fe_2_O_3_ nanoparticles via the sol–gel process [19]:2(FeCl_3_·6H_2_O) + 4C_2_H_5_OH → 2(C_2_H_5_O-Fe-OC_2_H_5_) + 6HCl + 11H_2_O + O_2_↑ (Hydrolysis)(1)
C_2_H_5_O-Fe-OC_2_H_5_ + H_2_O → HO-Fe-OH + 2C_2_H_5_OH↑ (Alcohol condensation)(2)
HO-Fe-OH + HO-Fe-OH → HO-Fe-O-Fe-OH + H_2_O↑ (Polymerization)(3)
2HO-Fe-O-Fe-OH + O_2_ → Fe_2_O_3_ + 2H_2_O↑ (Oxidation)(4)

### 2.3. Effect of α-Fe_2_O_3_/ZnO Heterojunction for Gas Sensing Mechanism

An n–n heterojunction at the interface between α-Fe_2_O_3_ and ZnO caused band bending (Figure 5a). Electrons transfer easily from α-Fe_2_O_3_ to ZnO because the conduction band energy (E_C_) in α-Fe_2_O_3_ is higher than that in ZnO. However, because the potential barrier at the interface forms because of the depletion layer, it is difficult to transfer electrons from α-Fe_2_O_3_ to ZnO. When exposed to the reducing gas, the potential barrier decreases, and electrons can transfer from α-Fe_2_O_3_ to ZnO [20,21]. Therefore, the change in the resistance increases owing to the reaction with a reducing gas, thus improving the sensitivity (Figure 5b).

## 3. Results and Discussion

### 3.1. Materials Analysis

#### 3.1.1. SEM Analysis

The surface of the ZnO nanostructure gas sensor was analyzed using field-emission scanning electron microscopy (FE–SEM). Figure 6a illustrates the ZnO NWs using only Zn powder. Figure 6b shows that the tip of the wire grown is rounded, which is evidence of NWs grown via VLS [22]. The NW vines grew longer and thinner than the NWs grown without adding Sn powder (Figure 6c). Consequently, the surface-area-to-volume ratio was larger in the NWs grown with Sn; thus, their sensitivity was higher. Figure 6d shows the nanostructures with added 0.1 M α-Fe_2_O_3_, which is not shaped as the NW vines in Figure 6c. The surface-area-to-volume ratio of the nanostructure decreased, which affected the response negatively. The NW vines decorated with 0.05-M α-Fe_2_O_3_ via the sol–gel process maintained a vine shape (Figure 6e). These ZnO NW vines were decorated with α-Fe_2_O_3_ nanoparticles, as shown in Figure 6f.

#### 3.1.2. EDS and XRD Analysis

NW vines decorated with 0.05 M α-Fe_2_O_3_ were analyzed using energy dispersive X-ray spectroscopy (EDS) (Figure 7). The NW vines were grown by adding Sn at a weight ratio of 1:1 (Sn:Zn); however, the EDS analysis indicates that they contained 6.7 wt% Sn and 65.5 wt% Zn. Thus, Sn was the catalyst for the growth of ZnO NWs, as shown in the results in Figure 6a,c above. The value was similar to the theoretical value. Therefore, Zn and Fe might exist as ZnO and α-Fe_2_O_3_ on the device.

The NW vines decorated with 0.05 M α-Fe_2_O_3_ were analyzed using X-ray diffraction (XRD). The XRD peak pattern from the ZnO NW vines decorated with 0.05-M α-Fe_2_O_3_ (Figure 8a) contained peaks corresponding to ZnO (Figure 8b) and α-Fe_2_O_3_ (Figure 8c). The XRD pattern was indexed to hexagonal ZnO (Ref. Code 01-070-2551) and hematite α-Fe_2_O_3_ (ICSD no. 201096). This confirms that ZnO and α-Fe_2_O_3_ existed on the gas sensor. 

### 3.2. Gas Sensing Measurements

All the gas sensing measurements were performed by exposing the reaction gas to a stabilized gas sensor for 5 min, removing the reaction gas, and stabilizing for 10 min. The resistance changes measured from three different sensors after exposure to 100 ppm CO_2_ and NH_3_ at 300 °C are shown in Figure 9 and Table 1. All sensors indicated the characteristics of an n-type semiconductor-type gas sensor. When Sn was added, the surface-to-volume ratio increased, and the response was improved. When α-Fe_2_O_3_ was deposited, the thickness of the electron depletion layer increased, and the response was improved.

The response is defined as the resistance in air, R_a_, divided by the resistance in the reactive gas, R_g_. The responses of the NW vines to 100 ppm CO, NH_3_, and NO_2_ at various molar concentrations of decorated α-Fe_2_O_3_ nanoparticles at 300 °C are shown in Figure 10 and Table 2. The response to NO_2_, which is an oxidative gas, decreased as the molar concentration of α-Fe_2_O_3_ increased. The responses to CO were stronger than those to NH_3_ at all molar concentrations. When the vines were decorated with 0.025 M α-Fe_2_O_3_, the response to both gases was similar to that of the NW vines without α-Fe_2_O_3_ because the α-Fe_2_O_3_ content might be too small to produce any effect. The response of the NW vines decorated with 0.05 M α-Fe_2_O_3_ was improved significantly. The response to both gases decreased as the molar concentration of α-Fe_2_O_3_ increased above 0.05 M. An excess concentration of α-Fe_2_O_3_ destroyed the NW vines, as shown in the FE–SEM images. This implies that the surface area of the gas sensor decreased, and hence the response decreased as the molar concentration of α-Fe_2_O_3_ increased above 0.05 M.

The responses to 100 ppm CO, NH_3_, and NO_2_ of at temperatures ranging from 100 to 350 °C in 50 °C increments are shown in Figure 11a and Table 3. Likewise, the response to CO was higher than those to NH_3_ and NO_2_ at all temperatures. The highest response to these gases was observed and the difference between the responses to three gases was the greatest at 300 °C. In Figure 10 and Figure 11, although the response to NH_3_ improved by the addition of α-Fe_2_O_3_, the response to CO remained higher. Figure 11b shows the resistance change of the gas sensors at 100 °C.

The response of the NW vines decorated with 0.05 M α-Fe_2_O_3_ at concentrations of CO ranging from 1 to 100 ppm was measured at 300 °C. R_g_ became lower as the concentration of CO gas increased from 1 to 100 ppm, as shown in Figure 12a and Table 4. Further, the response stability of the sensor is highly important for its practical application. The relationship between the response to CO and CO concentration was nearly linear (coefficient of determination, R^2^ = 0.99), as shown in Figure 12b. The NW vines decorated with 0.05 M α-Fe_2_O_3_ maintained a high sensing response to 100 ppm CO during the sensing stability test, as shown in Figure 12c.

## 4. Conclusions

We fabricated α-Fe_2_O_3_/ZnO n–n heterojunction nanostructures via VLS growth and the sol–gel process. The gas sensing characteristics were investigated and the material was analyzed. When exposed to a reducing gas, electrons in the absorbed oxygen ions and α-Fe_2_O_3_ nanoparticles were transferred into ZnO NWs. The resistance of the α-Fe_2_O_3_/ZnO n–n heterojunction nanostructures decreased more than that of the undecorated ZnO NWs.

## Figures and Tables

**Figure 1 sensors-19-01903-f001:**
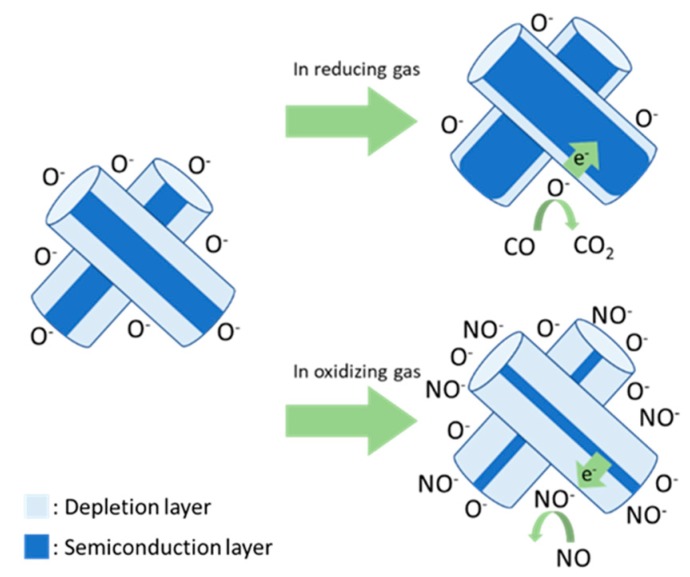
Schematic model of n-type semiconductor gas sensors.

**Figure 2 sensors-19-01903-f002:**
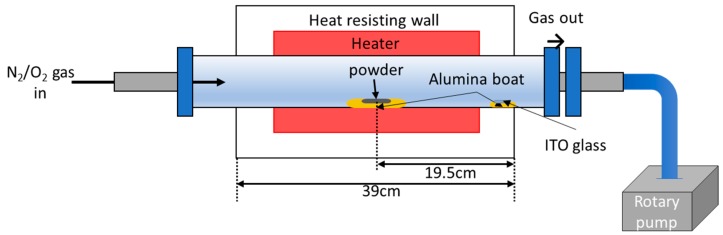
Schematics of the furnace and experimental equipment.

**Figure 3 sensors-19-01903-f003:**
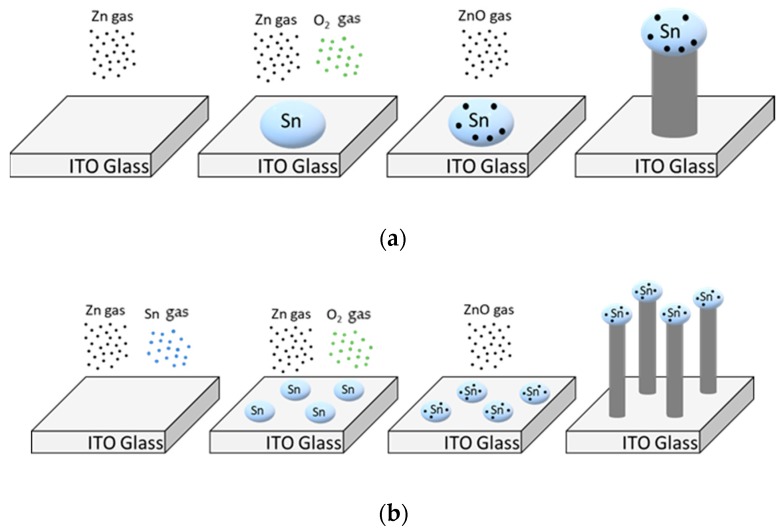
Schematics of the vapor–liquid–solid process using (**a**) only Zn powder and (**b**) Zn and Sn powder on ITO.

**Figure 4 sensors-19-01903-f004:**
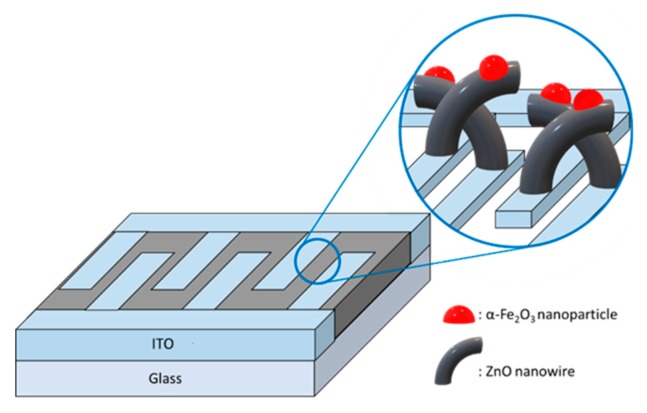
Schematics of ZnO nanowire (NW) vines decorated with α-Fe_2_O_3_.

**Figure 5 sensors-19-01903-f005:**
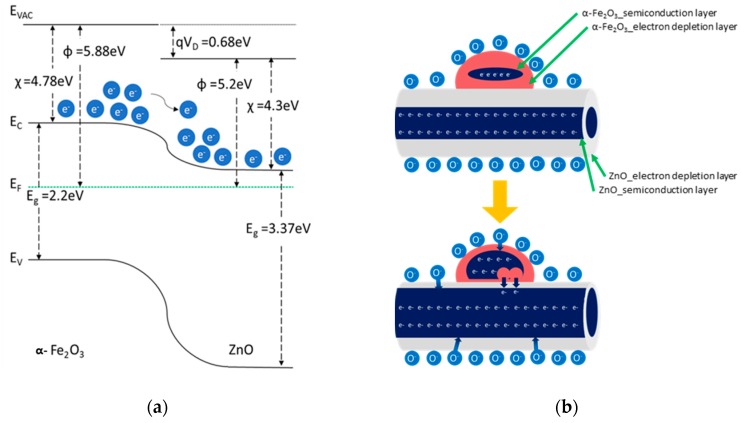
Improvement in gas sensor response: (**a**) energy diagram of α-Fe_2_O_3_/ZnO heterojunction and (**b**) schematics of α-Fe_2_O_3_/ZnO when exposed to a reducing gas.

**Figure 6 sensors-19-01903-f006:**
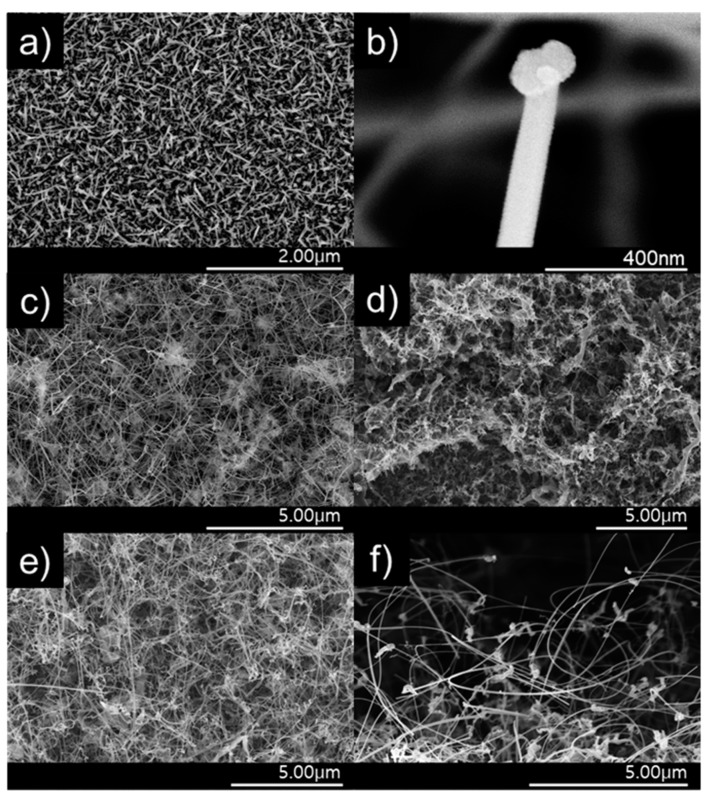
Field emission scanning electron microscopy (FE-SEM) images of (**a**) ZnO NWs, (**b**) the tip of an NW, (**c**) NW vines, (**d**) NW vines decorated with 0.1 M α-Fe_2_O_3_, (**e**) NW vines decorated with 0.05 M α-Fe_2_O_3_, and (**f**) side view of the NW vines decorated with 0.05 M α-Fe_2_O_3_.

**Figure 7 sensors-19-01903-f007:**
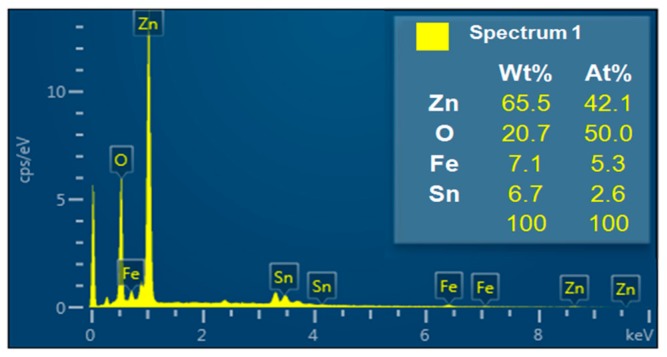
Energy dispersive X-ray spectroscopy (EDS) analysis of ZnO NW vines decorated with 0.05-M α-Fe_2_O_3_ on ITO.

**Figure 8 sensors-19-01903-f008:**
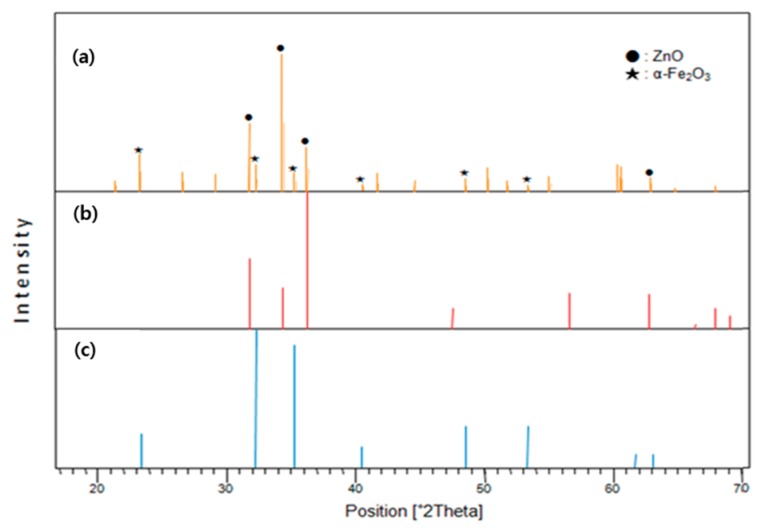
X-ray diffraction (XRD) analyses of (**a**) ZnO NW vines decorated with 0.05-M α-Fe_2_O_3_ on ITO, (**b**) hexagonal ZnO, (**c**) hematite α-Fe_2_O_3_.

**Figure 9 sensors-19-01903-f009:**
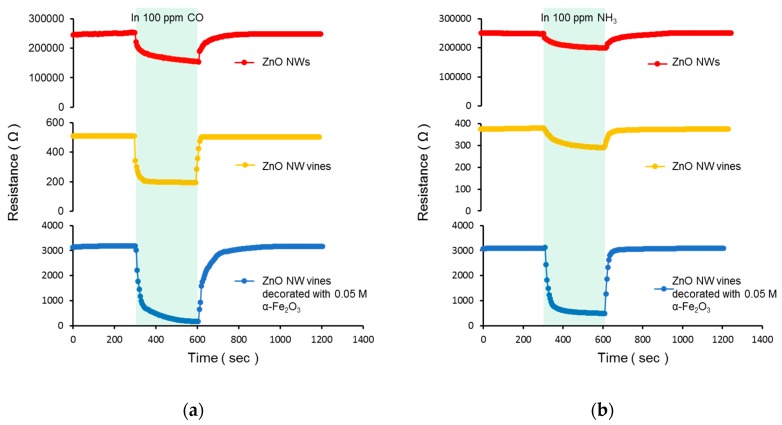
Resistance changes in ZnO NWs, ZnO NW vines, and ZnO NW vines decorated with 0.05 M α-Fe_2_O_3_ to (**a**) 100 ppm CO and (**b**) 100 ppm NH_3_.

**Figure 10 sensors-19-01903-f010:**
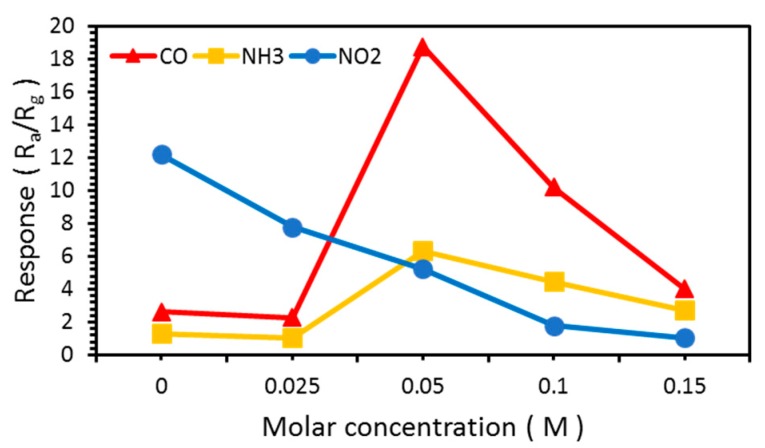
Response of ZnO NW vines to 100 ppm CO and NH_3_ at various concentrations of α-Fe_2_O_3_ at 300 °C.

**Figure 11 sensors-19-01903-f011:**
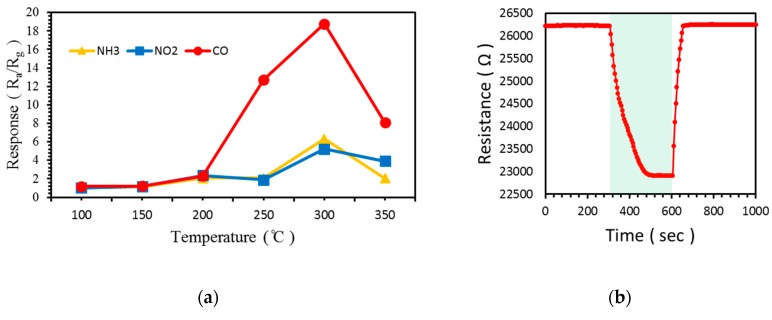
Response of ZnO NW vines decorated with 0.05 M α-Fe_2_O_3_ to 100 ppm CO, NH_3_, and NO_2_: (**a**) according to temperature from 100 to 350 °C, (**b**) at 100 °C.

**Figure 12 sensors-19-01903-f012:**
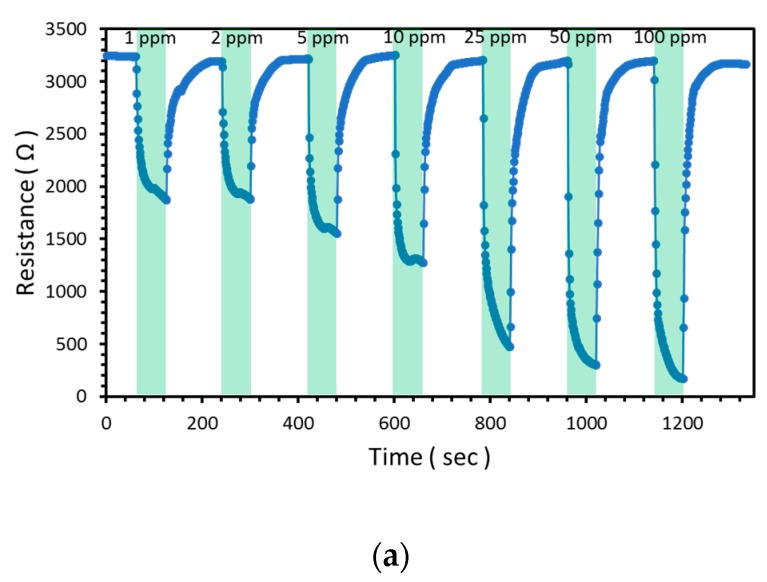
(**a**) Resistance change and (**b**) response of ZnO NW vines decorated with 0.05 M α-Fe_2_O_3_ to CO at concentrations ranging from 1 to 100 ppm at 300 °C, and (**c**) response stability of ZnO NW vines decorated with 0.05 M α-Fe_2_O_3_ toward 100ppm CO at 300 °C.

**Table 1 sensors-19-01903-t001:** Resistance of the ZnO NWs, ZnO NW vines, and ZnO NW vines decorated with 0.05-M α-Fe_2_O_3_ to 100 ppm CO or NH_3._

Molar Concentration (M)		ZnO NWs	ZnO NW Vines	ZnO NW Vines Decorated with 0.05 M α-Fe_2_O_3_
100 ppm CO gas	R_a_ (Ω)	250,000	500	3200
	R_g_ (Ω)	150,000	190	170
100 ppm NH_3_ gas	R_a_ (Ω)	250,000	370	3100
	R_g_ (Ω)	200,000	290	480

**Table 2 sensors-19-01903-t002:** Response at various concentrations of α-Fe_2_O_3_ at 300 °C.

Molar Concentration (M)	0	0.025	0.05	0.1	0.15
100 ppm CO gas	2.6	2.3	18.8	10.2	4.0
100 ppm NH_3_ gas	1.3	1.04	6.3	4.5	2.7
100 ppm NO_2_ gas	12.2	7.8	5.2	1.8	1.1

**Table 3 sensors-19-01903-t003:** Response of ZnO NW vines decorated with 0.05 M α-Fe_2_O_3_ at various temperatures.

Temperature (°C)	100	150	200	250	300	350
100 ppm CO gas	1.2	1.2	2.3	12.7	18.8	8.1
100 ppm NH_3_ gas	1.0	1.1	2.1	2.1	6.3	2.1
100 ppm NO_2_ gas	1.0	1.2	2.2	1.9	5.2	3.9

**Table 4 sensors-19-01903-t004:** Resistances and response of 0.05 M α-Fe_2_O_3_/ZnO NW vines according to gas concentration.

Gas Concentration (ppm)	1	2	5	10	25	50	100
R_a_ (Ω)	3239	3163	3208	3230	3179	3165	3176
R_g_ (Ω)	1870.5	1877.0	1551.7	1271.3	469.5	298.8	169.2
Response	1.7	1.7	2.1	2.5	6.8	10.6	18.8

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
