# Peer review of "Improved Sensitivity of α-Fe_2_O_3_ Nanoparticle-Decorated ZnO Nanowire Gas Sensor for CO"

_sensors, 2019, doi:10.3390/s19081903_

Reviewer 1 Report

The paper reports an interesting results of the ZnO nanowire based sensors.

However, I think that the data lacked some chemical approaches for sensor chemistry. So, additional discussions and comments should be add before publication.

Individual comments are as follows.

1)           How did you assign ZnO nanowire or ZnO nano vine sensors?

2)           Fig. 7; You measured the EDS for the device. So, why did you not show the EDS mapping?

3)           Fig. 8; I cannot understand differences between a, b and c in Fig. 8 even though I read the text.

4)           Fig. 9; CO for ZnO NW vines +FeOx, In 100 ppm CO the resistance of the device become to zero?

5)           Fig.11; In (b) you measured response at 100ºC, then why there were no data at 100ºC in Fig 11 (a) and Table 2.

6)           I believe that the cross sensitivity is one of the most important points for the gas sensors.

So, how about the response of this sensor to CO2, NOx, CxHx, etc.

In Fig. 12 (b), NH3 response should be add to this fig.

7)           Table 3; You should show the real value for Ra (ohm).

Author Response

Please see the attached file for the answers to reviewer's comments.

Reviewer 2 Report

This is well written report on gas sensing material with some explanation about gas sensing mechanisms. In my opinion it may be published in the present form.

Author Response

Thank you very much for your review.

Reviewer 3 Report

Comments to the Author

Authors present the preparation and gas sensing study of ɑ-Fe2O3 nanoparticles decorated ZnO nanowires, and the obtained results are very interesting. However, this manuscript is not well organized and there are lack of experiment data which seriously affect the integrity of the manuscript. Specific suggestions are offered below:

1. There are lack of material characterization in the Materials and Methods section, and the gas sensing mechanism should be discussed in more detail and put in the last part of the result and discussion section.

2. The selectivity of the sensor should be tested by exposing the sensor to different kinds of gases with defined concentration to compare their response to different gases.

3. The response repeatability and long term stability are vital for the application of gas sensors. Therefore, the repeatability and stability should be tested. The references: Journal of Materials Chemistry A, 5(21), 10387-10397 and Industrial & Engineering Chemistry Research, 2016, 55, 3588-3595 may be useful to you.

4. The SEM images should be tailored, the XRD curves of the samples instead of peak lines should be given out. The quality of the figures should be improved. ZnO NW vines are mentioned in the manuscript but their preparation process is not clearly described.

5. The sensing performance of the sample should be compared with other reported ZnO based materials.

6. The English of the manuscript needs polishing.

Author Response

Please see the attached file for the answers to reviewer's comments.

Round  2

Reviewer 1 Report

The MS was revised as I requested.

Then, I would like to recommend for publication, now. 

Reviewer 3 Report

The authors did a good job in reviewing this manuscript. The main concerns raised were satisfactorily addressed with additional experiments and extra discussion. Therefore, I consider that this manuscript is appropriate to be acceptable.